# Cryo-EM structure of a $Ca^{2+}$-bound photosynthetic LH1-RC complex containing multiple αβ-polypeptides

Kazutoshi Tani [1,9,10 ✉], Ryo Kanno[2,9], Yuki Makino[3], Malgorzata Hall[2], Mizuki Takenouchi[3], Michie Imanishi[4], Long-Jiang Yu [5], Jörg Overmann [6,7], Michael T. Madigan[8], Yukihiro Kimura[4], Akira Mizoguchi[1], Bruno M. Humbel[2] & Zheng-Yu Wang-Otomo [3,10 ✉]

The light-harvesting-reaction center complex (LH1-RC) from the purple phototrophic bacterium *Thiorhodovibrio* strain 970 exhibits an LH1 absorption maximum at 960 nm, the most red-shifted absorption for any bacteriochlorophyll (BChl) *a*-containing species. Here we present a cryo-EM structure of the strain 970 LH1-RC complex at 2.82 Å resolution. The LH1 forms a closed ring structure composed of sixteen pairs of the αβ-polypeptides. Sixteen Ca ions are present in the LH1 C-terminal domain and are coordinated by residues from the αβ-polypeptides that are hydrogen-bonded to BChl *a*. The $Ca^{2+}$-facilitated hydrogen-bonding network forms the structural basis of the unusual LH1 redshift. The structure also revealed the arrangement of multiple forms of α- and β-polypeptides in an individual LH1 ring. Such organization indicates a mechanism of interplay between the expression and assembly of the LH1 complex that is regulated through interactions with the RC subunits inside.

[1] Graduate School of Medicine, Mie University, Tsu 514-8507, Japan. [2] Imaging Section, Research Support Division, Okinawa Institute of Science and Technology Graduate University (OIST), 1919-1, Tancha, Onna-son, Kunigami-gun, Okinawa 904-0495, Japan. [3] Faculty of Science, Ibaraki University, Mito 310-8512, Japan. [4] Department of Agrobioscience, Graduate School of Agriculture, Kobe University, Nada, Kobe 657-8501, Japan. [5] Photosynthesis Research Center, Key Laboratory of Photobiology, Institute of Botany, Chinese Academy of Sciences, Beijing 100093, China. [6] Leibniz-Institute DSMZ-German Collection of Microorganisms and Cell Cultures, 38124 Braunschweig, Germany. [7] Faculty of Life Science, Institute of Microbiology, Braunschweig University of Technology, Braunschweig, Germany. [8] Department of Microbiology, Southern Illinois University, Carbondale, IL 62901, USA. [9] These authors contributed equally: Kazutoshi Tani, Ryo Kanno. [10] These authors jointly supervised this work: Kazutoshi Tani, Zheng-Yu Wang-Otomo. ✉email: ktani@doc.medic.mie-u. ac.jp; wang@ml.ibaraki.ac.jp

Photosynthetic organisms have evolved a diversity of light-harvesting strategies to absorb specific portions of the electromagnetic spectrum and thereby enable their growth in different habitats. Whereas oxygenic phototrophs synthesize chlorophylls as the major pigments that have long-wavelength absorption bands ($Q_y$ transition) in the relatively narrow range of 680–750 nm in vivo, the evolutionarily more ancient anoxygenic phototrophic bacteria produce a suite of bacteriochlorophylls (BChl) that extend their light absorption range from as low as 700 nm to beyond 1000 nm[1–3]. In purple phototrophic bacteria, the core light-harvesting complexes (LH1) usually display absorption maxima ~870 nm for BChl $a$-containing species and 1020 nm for the BChl $b$-containing species. However, among the BChl $a$-containing phototrophs, several species exhibit a red-shifted LH1-$Q_y$ to a range of 890–963 nm[4–7]. These organisms represent model systems for revealing the mechanism that phototrophic purple bacteria have evolved to avoid competition with other phototrophic species while at the same time avoiding spectral overlap with water molecules that absorb strongly near 980 nm.

The most thoroughly investigated LH1 complex showing a red-shifted $Q_y$ band is that from the thermophilic purple sulfur bacterium *Thermochromatium* (*Tch.*) *tepidum*, a phototroph isolated from a hot spring microbial mat in Yellowstone National Park (USA) and capable of growth up to 57 °C[4]. The *Tch. tepidum* LH1 complex, purified in a reaction center (RC)-associated form, revealed a $Q_y$ at 915 nm and enhanced thermostability compared with those from mesophilic counterparts[8]. Both the LH1-$Q_y$ redshift and increased thermostability of the *Tch. tepidum* LH1-RC are due to the binding of calcium ions to the LH1 polypeptides[9–11]. Crystal structures of the *Tch. tepidum* LH1-RC complex confirmed these findings and identified sixteen Ca-binding sites in the LH1 complex near the BChl $a$ coordination sites[12]. Each $Ca^{2+}$ is ligated by three residues from an $\alpha$-polypeptide, one from a $\beta$-polypeptide, and two water molecules, forming a double-pyramid coordination structure[13]. As a consequence, the LH1-$\alpha$ (inner) and $\beta$(outer) rings form a tight network connected by $Ca^{2+}$. This leads to a more rigid structure for the entire LH1 complex and highly restricts molecular motions around the BChl $a$-binding domain causing inhomogeneous narrowing in the spectroscopy[14,15]. Collectively, these structural features explain the observed LH1-$Q_y$ redshift and elevated thermostability of the *Tch. tepidum* LH1-RC complex.

Based on the work on *Tch. tepidum* LH1-RC, we have examined the LH1 complex from a spectrally unique mesophilic purple sulfur bacterium, *Thiorhodovibrio* (*Trv.*) strain 970; this phototroph shows an LH1-$Q_y$ at 960 nm (Supplementary Fig. 1), the most red-shifted $Q_y$ of all BChl $a$-possessing species[6]. As for the thermophilic *Tch. tepidum*, calcium ions were responsible for this ultra-redshift; removal of $Ca^{2+}$ resulted in a blue-shift of the LH1-$Q_y$ to 875 nm, whereas the subsequent addition of $Ca^{2+}$ restored 960-nm absorbance[16]. Although this process required $Ca^{2+}$ and was freely reversible, the structural basis behind these $Ca^{2+}$-induced spectral changes remained unsolved. In addition to the large redshift of the LH1-$Q_y$, the special pair BChl $a$ dimer in the RC of *Trv.* strain 970 also exhibits a red-shifted absorption band at 920 nm[6], the longest wavelength reported for all BChl $a$-containing RCs (which typically range from 865 nm to 890 nm). Although "uphill" energy transfer is common in bacterial photosynthesis[17], the energy gap (425 cm$^{-1}$) between LH1 and RC in *Trv.* strain 970 is the largest among those reported for wild-type purple bacteria (130–350 cm$^{-1}$) including the BChl $b$-containing species[6]. Despite the large uphill nature, a time constant of 65 ps was observed for the trapping of excitation energy from LH1 by the RC in *Trv.* strain 970, a value typical for other purple bacteria[6]. Elucidating the structural basis for this ultra-redshift of the

special pair would reveal how RCs have evolved to coordinate absorbance with their LH1 partner in order to maintain an appropriate energy gap for energy transfer.

We present here the cryo-EM structure of the LH1-RC from *Trv.* strain 970. Our results provide the first structural insights into the mechanisms that underlie the unique absorbance properties of both the LH1 and the RC of this spectrally unusual purple bacterium, mechanisms that may have relevance for other purple bacteria with red-shifted LH-RC complexes as well. The highly detailed density map of the *Trv.* strain 970 LH1-RC complex obtained in our work is the first to unravel the structural arrangement of multiple forms of $\alpha$- and $\beta$-polypeptides in the LH rings along with the multiple detergent-binding sites that exist in purified LH1 complexes.

## Results

**Structural overview**. The cryo-EM structure of *Trv.* strain 970 LH1-RC was determined at 2.82 Å resolution (Supplementary Table 1 and Supplementary Figs. 2–4), and its overall structure draws both parallels and contrasts with that of the *Tch. tepidum* LH1-RC[13]. The LH1 subunits of strain 970 are uniformly distributed around the RC forming a closed, slightly elliptical double cylinder composed of 16 pairs of helical $\alpha$(inner)$\beta$(outer)-polypeptides ($\alpha\beta$-dimer), 32 BChls $a$, 16 carotenoids (3,4,3′,4′-tetra-hydrospirilloxanthin)[16], and 16 Ca ions (Fig. 1 and Supplementary Figs. 5 and 6). The pigment and $Ca^{2+}$ stoichiometry are consistent with those determined by biochemical analysis[16]. The long and short dimensions of the outer LH1 $\beta$-polypeptide ring are 114 Å and 107 Å (distances between the outer edges of opposite helices), respectively. The RC is accommodated in the LH1 ellipsoid and fits the shape of the inner LH1 $\alpha$-ring with the L- and M-subunits in close proximity to the LH1 $\alpha$2- and $\alpha$4-polypeptides, respectively, in the transmembrane region (Fig. 1b, Supplementary Table 2, and Supplementary Fig. 5). Structures of the *Trv.* strain 970 RC subunits are nearly identical to those of corresponding subunits in the *Tch. tepidum* RC with smaller root-mean-square deviations (RMSD) for the integral membrane protein subunits (L and M) and larger RMSD for the membrane-associated protein subunits (C and H; Supplementary Fig. 7). Cofactors in the RC include four BChls $a$, two bacteriopheophytins (BPhe) $a$, one 15-*cis*-carotenoid, a menaquinone (MQ)-8 at the $Q_A$ site, and a ubiquinone (UQ)-8 at the $Q_B$ site.

BChl $a$ molecules in the strain 970 LH1 are aligned with the RC special pair (BChl $a_L$ and BChl $a_M$) on the same plane that parallels the membrane surface (Fig. 1c). The bacteriochlorins of the special pair point approximately toward the middle of the long and short axes of the LH1 BChl $a$ ring (Fig. 1d). The non-heme iron and the head groups of MQ and UQ in the RC are aligned along a line parallel to the long axis of the LH1 BChl $a$ ring. The LH1 BChls $a$ are ligated by histidine residues ($\alpha$-His37 and $\beta$-His36, Supplementary Fig. 8) with markedly longer coordination lengths (Supplementary Table 3) compared with those of the light-harvesting complexes from other species. The 32 BChl $a$ molecules in LH1 form an elliptical and partially overlapping ring-shaped arrangement with average Mg–Mg distances of 9.0 Å within a dimer and 8.8 Å between dimers (Fig. 1d). Carotenoids in strain 970 LH1 spanned the transmembrane region between the $\alpha$- and $\beta$-polypeptides and showed no apparent association with the polypeptides or BChl $a$.

**The $Ca^{2+}$- and BChl $a$- binding sites**. Of the 16 Ca ions present in the *Trv.* strain 970 LH1 C-terminal region between the inner and outer rings (Fig. 2a), each is coordinated by the main chain oxygen atoms of $\alpha$-Trp47 and $\alpha$-Ile52, the sidechain carboxyl

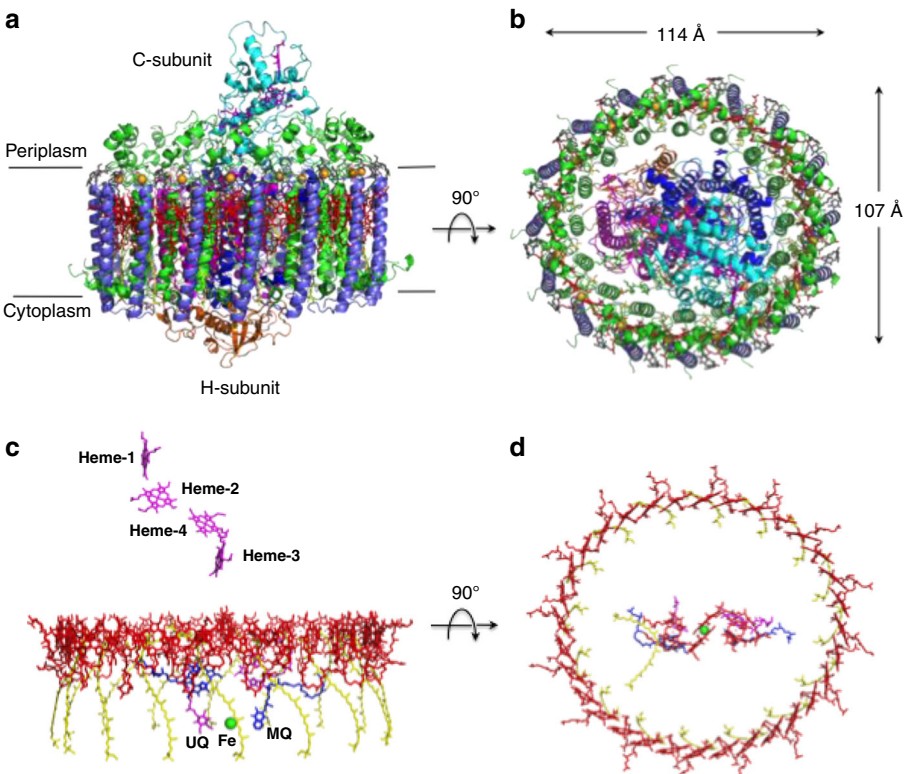

**Fig. 1 Overall structure and cofactor arrangement of the _Trv_. strain 970 LH1-RC complex. a** Side view of the LH1-RC parallel to the membrane plane with the periplasmic C-subunit above and the cytoplasmic H-subunit below. **b** Top view of the LH1-RC from the periplasmic side of the membrane. **c** Side view of the cofactor arrangement along the membrane plane with the periplasm above and the cytoplasm below. **d** Top view of the cofactor arrangement from the periplasmic side of the membrane. Hemes in the C-subunit are omitted for clarity. Color scheme: LH1 α, green; LH1 β, slate blue; L-subunit, magenta; M-subunit, blue; C-subunit, cyan; H-subunit, orange; $Ca^{2+}$, gold ball; BChl _a_, red sticks; 3,4,3′,4′-tetrahydrospirilloxanthin, yellow sticks; DDM, black sticks. Phospholipids are omitted for clarity. Cofactors in L- and M-subunits of RC are colored in magenta and blue, respectively.

group of α-Asp50, and the main chain oxygen atom of β-Trp45 in the adjacent subunit (Fig. 2b). Water molecules may also participate in the ligation, as occurs in the _Tch. tepidum_ LH1[13], although their densities were not identified. All of the $Ca^{2+}$-ligating residues are conserved in the expressed LH1 polypeptides (Fig. 2e and Supplementary Fig. 8). As a result, the Ca ions connect the α-inner and β-outer rings and therefore stabilize the entire LH1 structure. Because the $Ca^{2+}$-binding network is close to the LH1 BChl _a_ molecules, both static and dynamic properties of the coupled pigments are likely modified through the nearby $Ca^{2+}$-bound α-Trp47 and β-Trp45 residues whose sidechains are hydrogen-bonded to the C3-acetyl oxygen atoms of BChls _a_ (Fig. 2c, d). This $Ca^{2+}$-binding effect has been demonstrated to contribute to the large redshifts of the LH1-Qy transitions in both _Tch. tepidum_[9,14,18–20] and _Trv._ strain 970[16].

A unique feature of the _Trv._ strain 970 LH1 is that another histidine residue (His48) in the LH1 α-polypeptide, a residue absent from other purple bacterial LH1 α-polypeptides (Supplementary Fig. 8a)[21], also participates in hydrogen bonding with the C3-acetyl group of BChls _a_ (Fig. 2c) in addition to Trp47 that is conserved in all LH1 α-polypeptides. Furthermore, the His48 also shows π–π interaction with Trp47 (Fig. 2c). The additional hydrogen bond between His48 and BChl _a_, together with an interacting triangle formed with BChl _a_–Trp47–His48–BChl _a_, further strengthens interactions between the pigment and surrounding polypeptides and stabilizes the conformation of the BChl _a_ molecules. This structure is in good agreement with resonance Raman results on the _Trv._ strain 970 LH1-RC complex[16] that showed a markedly red-shifted C3-acetyl C=O stretching band and indicate even stronger hydrogen bonding for

the carbonyl oxygen than occurs in _Tch. tepidum_ LH1. This enhanced hydrogen bonding between BChl _a_ and polypeptides likely contributes significantly to the ultra-redshift of the LH1-Qy transition observed in _Trv._ strain 970.

**Arrangement of the LH1 multiple polypeptides.** The sequence of the _Trv._ strain 970 genome (GenBank assembly accession: GCA_000228725.3) revealed five and four putative genes encoding the LH1 α- and β-polypeptides, respectively (Supplementary Fig. 8). Expressions of four forms of the α-polypeptides (α1–4) and two forms of the β-polypeptides (β1 and β4) were identified in the purified LH1-RC by TOF/MS and HPLC (Supplementary Fig. 9). These β-polypeptides correspond to the gene products of $pufB_1$ and $pufB_2$ reported previously[21]. All of the expressed LH1 polypeptides were assigned in the cryo-EM structure (Fig. 3a, b and Supplementary Fig. 5). Six copies of the α1- and eight copies of the α3-polypeptides form two groups and align on the two sides of the LH1 ellipse. The α1- and α3-polypeptides only differ in the C-terminal regions where the α3-polypeptide is four residues longer than the α1-polypeptide (Supplementary Fig. 8b). The α3-polypeptides have weak interactions with the RC C-subunit through their slightly longer C-terminal domains, whose solvent-exposed entity is largely tilted toward the α3-crescent.

One α2- and one α4-polypeptide were identified in the cryo-EM structure (Fig. 3a, b and Supplementary Fig. 5b). These two polypeptides have longer C-terminal domains and a higher sequence similarity than other α pairs, except for α1/α3 (Supplementary Fig. 8c). The α2- and α4-polypeptides are located at unique positions between the α1- and α3-crescents and interact

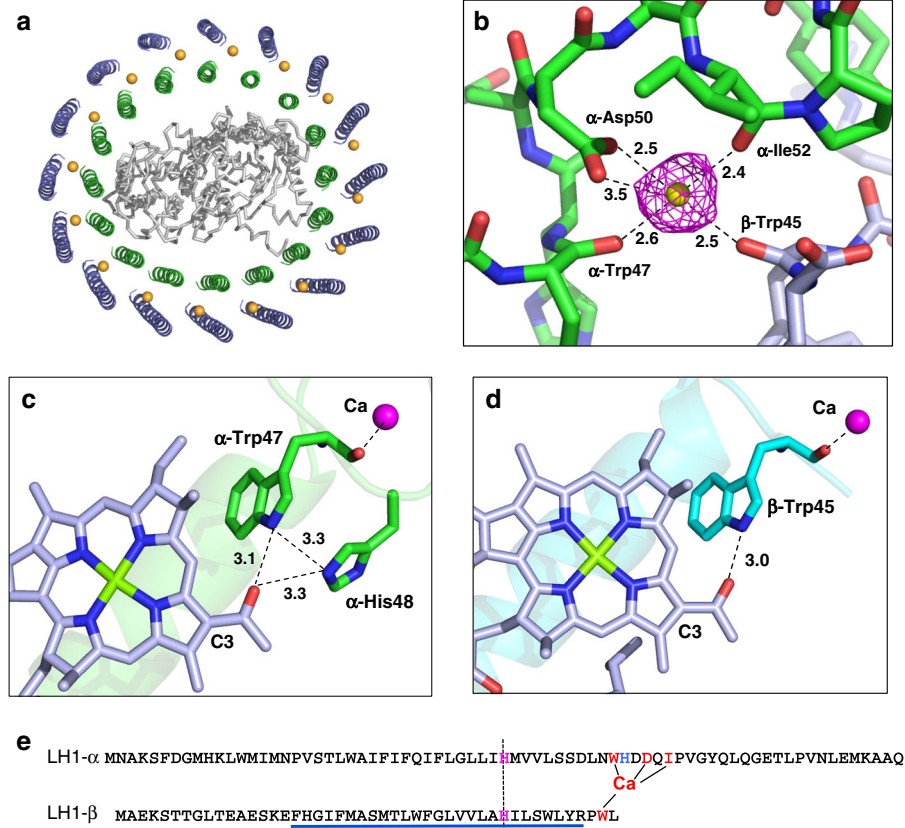

**Fig. 2 Ca²⁺- and BChl *a*-binding sites in the *Trv*. strain 970 LH1 complex. a** Top view of the LH1-RC from the periplasmic side of the membrane. Color scheme: LH1 α, green; LH1 β, slate blue; L- and M-subunits, gray; Ca²⁺, orange balls. **b** A typical Ca²⁺-binding site in LH1. Color scheme: LH1 α, green; LH1 β, light-blue; Ca²⁺, yellow ball. Density map is shown in magenta mesh around the Ca²⁺ at a contour level of 2.5σ. **c** Hydrogen bonding site around BChl *a* C3-acetyl group (light-blue sticks) in LH1 α-polypeptide (green). The Ca²⁺ (magenta ball) is ligated by the main chain oxygen atom of α-Trp47. **d** Hydrogen bonding site around BChl *a* C3-acetyl group (light-blue sticks) in LH1 β-polypeptide (cyan). The Ca²⁺ (magenta ball) is ligated by the main chain oxygen atom of β-Trp45. **e** Sequence scheme showing the relative positions of the Ca²⁺- and BChl *a*-binding sites. The LH1 αβ-polypeptides are aligned relative to the BChl *a*-coordinating histidine residues (magenta letters with vertical dashed line). Underlined region represents presumed membrane-spanning domain. All distances are in ångströms.

in the transmembrane domains with the RC L- and M-subunits, respectively, through sidechains of Arg residues (α2-Arg19, α2-Arg20, and α4-Arg20) that are specific for the α2- and α4-polypeptides. The solvent-exposed C-terminal domain of the α2-polypeptide forms a rectangular-shaped conformation on the periplasmic side of the membrane and is stabilized by interactions with surrounding RC proteins through the N- and C-terminal domains of the C-subunit and the C-terminal domain of the M-subunit (Fig. 3c and Supplementary Fig. 5b). The long C-terminal domain of the α4-polypeptide forms a large loop spanning the presumed periplasmic membrane surface and interacts with both RC C- and M-subunits (Fig. 3c and Supplementary Fig. 5b).

Three and thirteen copies of the LH1 β1- and β4-polypeptides, respectively, were assigned in the cryo-EM structure. The two β isomers have high sequence similarity with slight differences in the N-terminal domains. The three β1-polypeptides are located at triangle positions in the LH1 outer ring (Fig. 3a, b) with their N-terminal domains in the vicinities of the RC H- and M-subunits. This indicates that the subtle modifications in the N-terminal sequence of the β-polypeptides likely facilitate interactions with the RC subunits on the cytoplasmic membrane surface.

**Structural features for the special pair of RC.** In addition to the major redshift of the LH1 of *Trv*. strain 970, the special pair BChl *a* dimer in the RC of this phototroph displays a largely red-shifted

absorption band at 920 nm[6], the longest wavelength among all BChl *a*-containing RCs reported. The *Trv*. strain 970 RC structure from our study provides further insights into the structure-function relationships of this unusual feature. Most notably, the coordination lengths of the His residues to the special pair BChls *a* are markedly longer (Fig. 4a and Supplementary Table 3) than in the RCs of *Tch. tepidum*[13], *Rhodobacter* (*Rba.*) *sphaeroides*[22], and the BChl *b*-containing *Blastochloris* (*Blc.*) *viridis*[23]. This is likely due to relatively strong hydrogen bonds formed between the imidazole groups of the coordinating His residues and nearby main chain oxygen atoms.

Key residues that participate in hydrogen bonding with the L-subunit-bound BChl *a*ₗ, along with a partially overlapping Phe (L-Phe168), are all conserved in the L-subunits of the BChl *a*-containing *Trv*. strain 970, *Tch. tepidum* and *Rba. sphaeroides* (Fig. 4b and Supplementary Fig. 10). In contrast, differences exist in the residues interacting with the M-subunit-bound BChl *a*ₘ. In *Trv*. strain 970, the M-Tyr197 forms a hydrogen bond with the BChl *a*ₘ C3-acetyl group (Fig. 4c and Supplementary Fig. 10). This is also true of *Tch. tepidum* but differs from that of *Rba. sphaeroides*, where the corresponding residue is Phe, which does not form a hydrogen bond. The most striking two features are that (i) a partially BChl *a*ₘ-overlapping Phe (M-Phe196) exists only in the *Trv*. strain 970, whereas it is a Leu in the BChl *a*-containing *Tch. tepidum* and *Rba. sphaeroides*, and (ii) a Ser

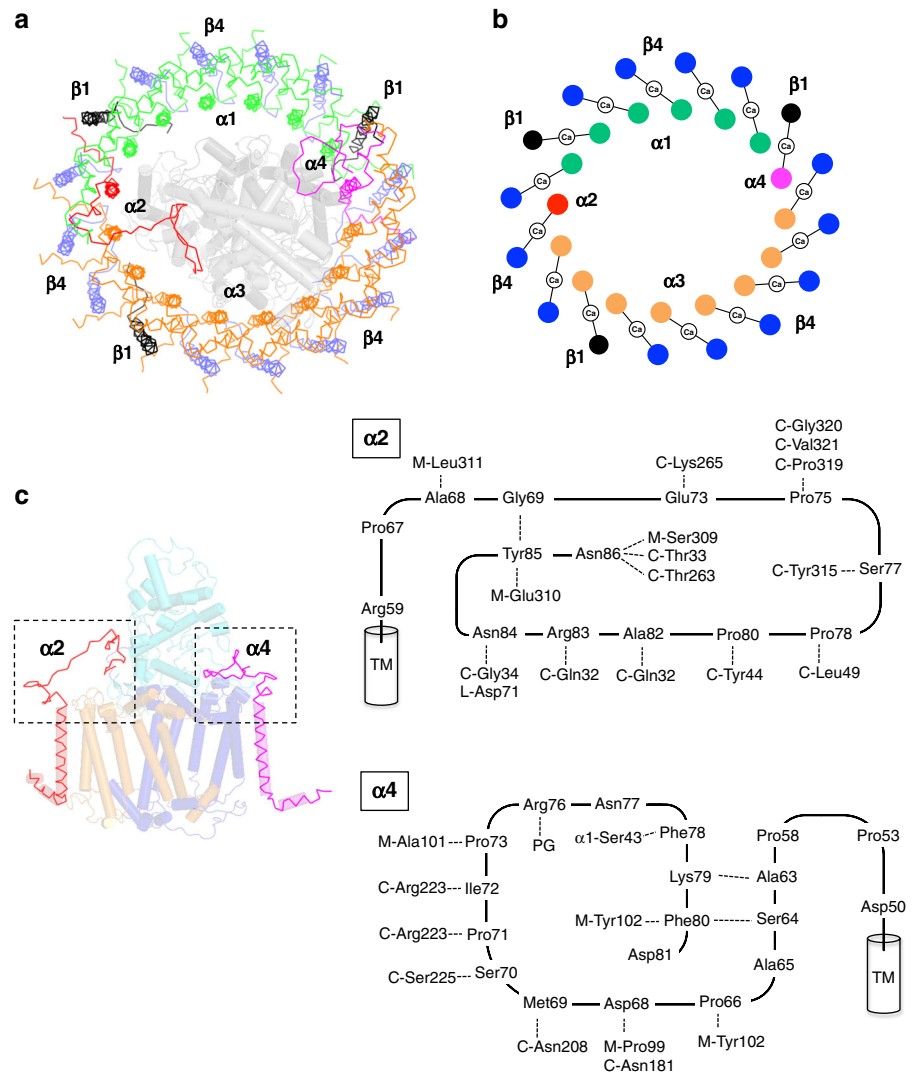

**Fig. 3 Arrangement of the LH1 multiple polypeptides. a** Top view of the LH1-RC from periplasmic side of the membrane. Color scheme: α1, green; α2, red; α3, orange; α4, magenta; β1, black; β4, blue. The RC L-, M-, and C-subunits are shown in gray cartoon. **b** Illustration of the arrangement of the LH1 multiple polypeptides bound with Ca ions. Same color scheme as in **a**. **c** Side view of the LH1 α2- and α4-polypeptides with RC L-, M-, and C-subunits shown by transparent orange, blue, and cyan cartoons, respectively (Note: the color schemes for L- and M-subunits are different from those in Fig. 1). The C-terminal domains of the two polypeptides marked by the dashed boxes are illustrated to show the interactions with surrounding proteins by dashed lines.

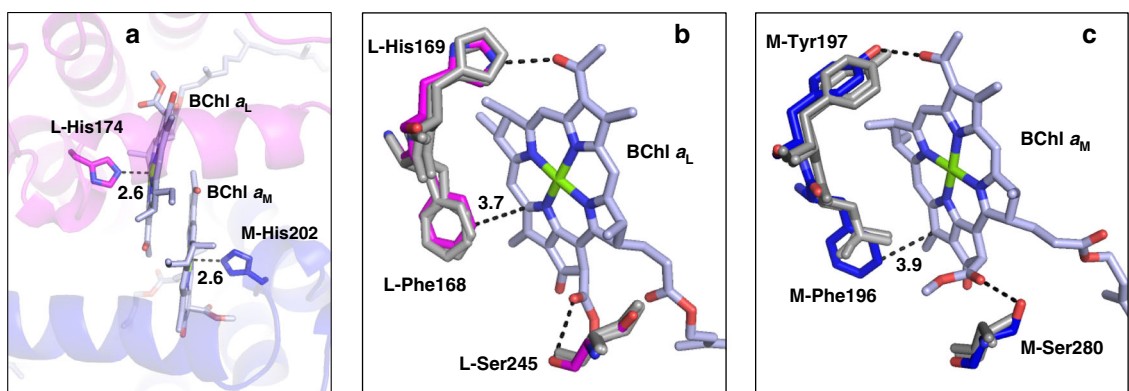

**Fig. 4 Key residues around the *Trv*. strain 970 RC special pair BChl *a* dimer. a** Top view of the BChl *a* dimer whose central Mg atoms are coordinated by the His residues in the L- and M-subunits. **b** Superposition of the residues in the RC L-subunits of *Trv*. strain 970 (magenta), *Tch. tepidum* (gray), and *Rba. sphaeroides* (gray) that form hydrogen bonds or interact with the BChl *a* molecule. **c** Superposition of the residues in the RC M-subunits of *Trv*. strain 970 (blue), *Tch. tepidum* (gray), and *Rba. sphaeroides* (gray) that form hydrogen bonds or interact with the BChl *a* molecule. All distances are in ångströms.

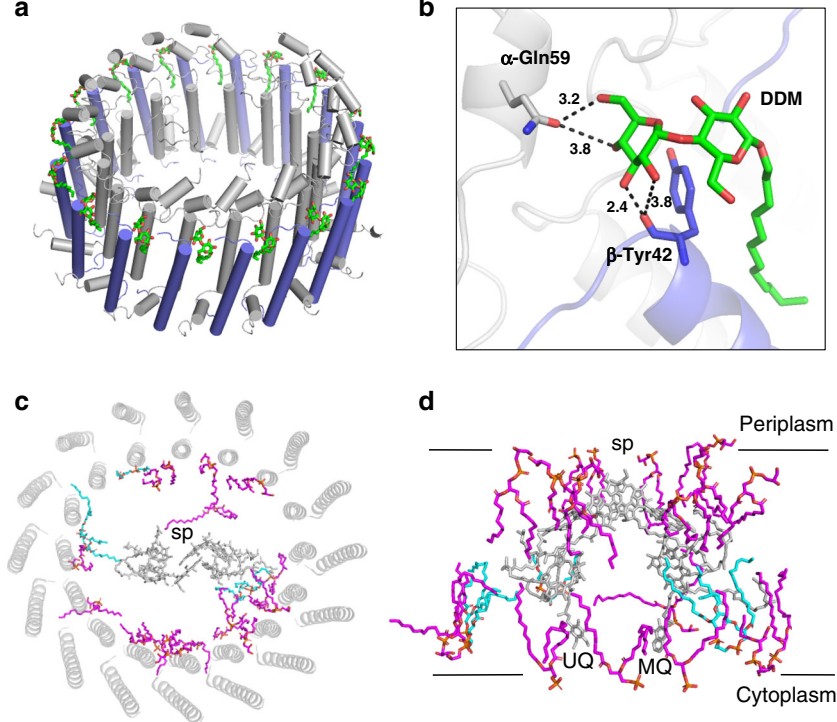

**Fig. 5 Detergents and phospholipids associated with the purified LH1-RC complex. a** Tentatively assigned DDM molecules (green sticks) in the LH1 outer ring. **b** Expanded view of a putative DDM-binding site showing the interactions with the residues in LH1 α- (gray) and β- (blue) polypeptides. All distances are in ångströms. **c** Phospholipid distribution in LH1-RC viewed from the periplasmic side of the membrane with the special pair (sp) positioned at the center. CL: cyan; PG: magenta. The pigment and quinone (UQ and MQ) molecules in the RC (hemes in the C-subunit were omitted for clarity) and the LH1 polypeptides are colored in gray. **d** Side view of the phospholipid distribution along the presumed membrane planes. Same color scheme as in **c**. The LH1 complex was omitted for clarity.

residue (M-Ser280) forms an additional hydrogen bond with the BChl $a_M$ C13²-ester group in strain 970, whereas the corresponding residue is Ala in *Tch. tepidum* and Gly in *Rba. sphaeroides* (Fig. 4c and Supplementary Fig. 10). It should be noted that a Phe in the M-subunit of the BChl *b*-containing *Blc. viridis* is also located at the same position of the M-Phe196 in *Trv.* strain 970. These residues have been demonstrated to affect the spectral behavior of the special pair, and the increased number of hydrogen bonds appears to contribute to the further redshifts of the *Trv.* strain 970 special pair absorption band (see Discussion section)[24–26].

**Detergents, lipids, and quinones**. Ordered densities were identified on the periplasmic side between the *Trv.* strain 970 LH1 β-polypeptides and can be best fit with molecules of the detergent DDM (Fig. 5a). The head groups of the DDM lie on the presumed membrane surface and interact with LH1 αβ-polypeptides (Fig. 5b) leading to stabilization of both DDM and the LH1 ring. The alkyl chains of the DDM are inserted into the transmembrane regions. The putative DDM molecules are located at similar positions to those of the LH1 γ-polypeptides of *Blc. viridis* (Supplementary Fig. 11)[23]. Residual densities at the same positions were also observed in the *Tch. tepidum* LH1-RC structure and were tentatively assigned with unknown alkyl chains[13].

³¹P-NMR measurement of the phospholipids extracted from purified *Trv.* strain 970 LH1-RC yielded approximately 47 phosphatidylglycerol (PG), 12 cardiolipin (CL), and 17 phosphatidylethanolamine (PE; Supplementary Fig. 12a). A total of 21 phospholipids (18 PG and 3 CL) were tentatively assigned in the cavities between the RC and LH1 based on the density map (Fig. 5c, d). The different head groups of PG and PE could not be distinguished at the current resolution. All of the CL molecules were oriented toward the cytoplasmic side of the membrane with

their head groups aligned on the membrane surface, whereas the PG molecules were distributed on both cytoplasmic and periplasmic sides. In addition, a diacylglycerol is covalently bound to the N-terminus of the RC C-subunit. These lipid distribution features are similar to those observed in the *Tch. tepidum* LH1-RC[13,27].

Biochemical analysis revealed approximately four UQ-8 and one MQ-8 in the purified *Trv.* strain 970 LH1-RC (Supplementary Fig. 12b). Only one UQ-8 could be assigned in the cryo-EM structure in addition to the UQ-8 at the Q_B site. The head group of the additional UQ-8 appears to interact with sidechains of the RC L-Gln88 and L-Thr191 in the transmembrane region. Channels exist between every adjacent pair of the LH1 αβ-polypeptides (Supplementary Fig. 13), as is also observed in the *Tch. tepidum* LH1-RC[12,13]. The channel openings nicely fit the size and shape of the ubiquinone head group, suggesting that these channels may serve as quinone paths. The putative ubiquinone channels seem to be mainly defined by the LH1 α-polypeptides because they form the narrowest portion of the openings with highly hydrophobic residues.

**Discussion**
Our structural determination of a Ca²⁺-bound LH1-RC from the purple sulfur phototrophic bacterium *Trv.* strain 970 here provides the second example of this particular arrangement, the first being that of *Tch. tepidum*[12,13]. However, the effect of Ca²⁺ on the redshift of the LH1-Q_y transition is more dramatic in *Trv.* strain 970 than in *Tch. tepidum*[16]. Much of the insight gained from the thoroughly investigated *Tch. tepidum* LH1-RC[9–11,14,15,18,20,28–30] has paved the way for understanding the structural basis of the spectroscopic behavior of the *Trv.* strain 970 LH1-RC because the Ca²⁺-binding sites are similar in the two complexes. Both static and dynamic properties on the pigment–pigment and pigment–protein

interactions are known to influence the excitation energy of the BChl *a* molecules[14,19,31–33]. Among the many factors, hydrogen bonding between BChl *a* and its surrounding proteins has a direct effect on the lowest-energy transition, and one of the most effective methods for detecting the hydrogen bonds is resonance Raman spectroscopy. There is a linear relationship between the redshift of the Raman band in the BChl *a* C3-acetyl stretching mode and the redshift in the LH absorption maximum[7,18,34,35]. The redshift of the Raman band observed at 1621 cm$^{-1}$ for the *Trv.* strain 970 LH1 is the largest among all LH1 complexes reported[16], indicating that an increased number of hydrogen bonds are present. In the *Trv.* strain 970 LH1 structure, the unique His residue (α-His48) present in all its LH1 α-polypeptides participates in hydrogen bonding with the BChl *a* C3-acetyl group and has π–π interaction with the Ca$^{2+}$-coordinating α-Trp47. This forms a Ca$^{2+}$-stabilized hydrogen bonding network (Figs. 2c and 3b) that tightly locks the LH1 inner and outer rings and contributes to the ultra-redshift of the *Trv.* strain 970 LH1-Q$_y$ transition. The markedly elongate coordination lengths of the His–BChl *a* in LH1 (Supplementary Table 3) may also influence the BChl *a* site energy (see discussion below for the RC special pair).

However, hydrogen bonding with pigments can explain only part of the extra redshifts of the LH1-Q$_y$ transition as seen in the *Tch. tepidum* LH1-RC[19] because structural disorder and the dynamic property of the BChl *a*-binding environment also need to be taken into account[9,14,15,28,29,36]. This is supported by the fact that the full C-terminal domains of all LH1 α-polypeptides can be traced from the current density map (2.82 Å resolution) observed for the *Trv.* strain 970 LH1-RC, even though the chain lengths of the LH1 α-polypeptides of this phototroph are much longer (11–24 residues) than those of *Tch. tepidum* (Supplementary Fig. 8a). In contrast, a number of the C-terminal residues of the *Tch. tepidum* LH1 α-polypeptides were invisible in the 1.9-Å resolution structure due to their weak densities[13]. This indicates that the *Trv.* strain 970 LH1 complex is structurally more homogeneous and less mobile than the *Tch. tepidum* LH1[14,30,36], especially as regards the C-terminal domains that are close to the BChl *a*-binding sites. By contrast, the *Trv.* strain 970 LH1 β-polypeptides form hydrogen bonds through their C-terminal carboxyl group with the β-Tyr42 in the adjacent chain, further strengthening the connectivity between the LH1 αβ-subunits. Overall, the combined features of relatively ordered structure and restricted molecular motion highlight the structural integrity of the *Trv.* strain 970 LH1 complex, and this is likely an important factor affecting the significant redshift of this phototroph's LH1-Q$_y$ transition.

The structure of the *Trv.* strain 970 LH1-RC has revealed the organization of multiple forms of a polypeptide in a single LH1 ring. Chemical cross-linking experiments[37] and single-molecule spectroscopy[38] have demonstrated that peripheral light-harvesting (LH2) complexes have a heterogeneous polypeptide composition in the individual LH2 rings. However, it remains a challenge for structural biology, such as crystallography or NMR, to detect the structural arrangement within a stochastically mixed complex composed of multiple polypeptides with very similar amino acid sequences. In our study, all of the four LH1 α-polypeptides expressed in *Trv.* strain 970 were identified from the density map. They form the LH1 inner ring and are arranged in a highly characteristic way that can be understood in terms of their interactions with the RC subunits: the six α1-polypeptides with the shortest chain length are aligned on one side of the LH1 ellipse without apparent interaction with the RC; the eight α3-polypeptides are aligned on the other side with their slightly longer C-terminal domains interacting with the solvent-exposed portion of the tilted RC C-subunit; the single α2- and α4-polypeptides, each of which is located between the α1- and α3-crescents, interact with the RC L- and M-subunits in the transmembrane regions, and also interact

with the RC C-subunit with their long C-terminal domains on the periplasmic side. These results suggest that a mechanism exists for expression and assembly of the LH1 complex that is regulated through interactions with the RC subunits inside the ring. It is unclear whether the results of our LH1 study are applicable to the LH2 complex that also contains multiple forms of αβ-polypeptides because LH2 complexes apparently lack any protein and pigment inside their ring structure.

The ultra Q$_y$-redshift of the special pair BChl *a* dimer in the *Trv.* strain 970 RC may be explained by a combination of at least three structural features: (i) the elongate coordination lengths of the His residues to BChls *a*, (ii) an aromatic residue (Phe196) in the RC M-subunit that is in close proximity to BChl *a*$_M$, and (iii) an additional hydrogen bond formed between the Ser280 in the RC M-subunit and the BChl *a*$_M$. The relatively longer coordination lengths of the His–BChl *a* in the strain 970 RC special pair (Supplementary Table 3) indicate a reduced degree of electron donation from the His imidazole group to the BChl *a* macrocycles, leading to a relatively electron-deficient (more positively charged) state in the BChl *a* dimer. This likely contributes to its red-shifted absorption because it is known that the oxidized special pair dimer shows a largely red-shifted absorbance at 1250 nm[26,39,40].

In the *Trv.* strain 970 RC, an aromatic residue (Phe196) in the M-subunit is in close proximity to BChl *a*$_M$, and another aromatic residue is conserved at the symmetry-related position in the L-subunit (L-168, Fig. 4b); Phe for the BChl *a*-containing RCs and Trp for the BChl *b*-containing RC (Supplementary Fig. 10). The aromatic plane (L-Phe168) and the bacteriochlorin plane are partially overlapped with ~25° inclined toward each other, and therefore have π–π interaction. Mutation of the Phe to Leu in the *Rba. sphaeroides* RC L-subunit resulted in a 10-nm blue-shift of the Q$_y$-band of the special pair along with an increased initial electron transfer time constant, decreased charge recombination time constant, and an increased redox potential of the RC[25], indicating significant roles for this aromatic residue. Based on this result, a redshift of the special pair Q$_y$-band would be expected by changing the Leu196 in the RC M-subunits of *Tch. tepidum* and *Rba. sphaeroides* to Phe as present in *Trv.* strain 970. The Phe196 aromatic plane of *Trv.* strain 970 is inclined by about 34° toward the bacteriochlorin plane (Fig. 4c), indicating a possible π–π interaction. It is of interest to point out that a Phe residue exists at the corresponding position with similar conformation in the M-subunit of the BChl *b*-containing *Blc. viridis* (Supplementary Fig. 10) whose special pair shows an absorption band at 955 nm[41]. It is unique for the *Trv.* strain 970 to have an additional hydrogen bond formed between the Ser280 in the RC M-subunit and the BChl *a*$_M$ C13$^2$-ester group (Fig. 4c). Introducing additional hydrogen bonds to the BChl *a* ring V of the special pair was demonstrated to result in a 10-nm redshift of the *Rba. sphaeroides* Q$_y$-band at 20 K along with slower forward electron transfer rates, faster charge recombination rates, and increased redox potentials of the RC[24]. All of the structural features of the *Trv.* strain 970 RC reveal a more symmetrical configuration along the pseudo-C2 symmetry axis, a characteristic more similar to that of the *Blc. viridis* RC than the RCs of *Tch. tepidum* and *Rba. sphaeroides* in terms of how key interacting residues are arranged around the special pair of BChl molecules. These features contribute not only to the ultra-redshifted Q$_y$-band of the *Trv.* strain 970 RC but also to its electron transfer rate, charge recombination rate, and RC redox potential.

In summary, our cryo-EM structure of the *Trv.* strain 970 LH1-RC provides a structural foundation for the unusual spectroscopic properties of this unique phototroph's LH1 and RC complexes. The characteristic organization of the multiple forms of LH1 polypeptides in an individual LH1 ring reveals a mechanism for the expression and assembly of the LH1 complex that is strictly

regulated through extensive interactions with the RC core inside the ring. Moreover, our study highlights the dramatic diversity of light-harvesting strategies in bacterial photosynthesis and a fine-tuned mechanism that has evolved for the sustainable growth of photosynthetic organisms in a changing environment.

## Methods

**Preparation and characterization of the LH1-RC complex.** The *Trv.* strain 970 cells were cultivated phototrophically (anoxic/light) at 22 °C for 7 days under incandescent light (60 W). Preparation of the *Trv.* strain 970 LH1-RC followed the previous procedure[16] with minor modifications. The chromatophores were suspended in a buffer containing 20 mM Tris-HCl (pH 8.5) and 10 mM $CaCl_2$ and were treated with 0.7% *n*-dodecyl β-D-maltopyranoside (DDM) at a concentration of $OD_{963} = 30$ to extract the LH1-RC. The crude LH1-RC was loaded on a DEAE column (Toyopearl 650 S, TOSOH) equilibrated with 20 mM Tris-HCl (pH 7.5) and 0.05% DDM at 7 °C. LH1-RC components were eluted by a linear gradient of $CaCl_2$ from 20 mM to 100 mM, and fractions with $A_{959}/A_{280} > 1.8$ were collected for the subsequent measurements (Supplementary Fig. 1), then assessed by negative-stain EM using a JEM-1011 instrument (JEOL). Masses and composition of the LH1 polypeptides were measured by matrix-assisted laser desorption/ionization time-of-flight mass spectroscopy (MALDI-TOF/MS) and reversed-phase HPLC, respectively, using methods described elsewhere[42]. Phospholipid and quinone contents in the purified LH1-RC were analyzed (Supplementary Fig. 10) as previously describe[27,43].

**Cryo-EM data collection.** Proteins for cryo-EM were concentrated to ~3 mg/ml. Three microliters of the protein solution were applied on a glow-discharged holey carbon grids (200 mesh Quantifoil R2/2 molybdenum), which had been treated with $H_2$ and $O_2$ mixtures in a Solarus plasma cleaner (Gatan, Pleasanton, USA) for 30 s and then blotted, and plunged into liquid ethane at −182 °C using an EM GP2 plunger (Leica, Microsystems, Vienna, Austria). The applied parameters were a blotting time of 6 s at 80% humidity and 4 °C. Data were collected at OIST on a Titan Krios (Thermo Fisher Scientific, Hillsboro, USA) electron microscope at 300 kV equipped with a Falcon 3 camera (Thermo Fisher Scientific). Movies were recorded using EPU software (Thermo Fisher Scientific) at a nominal magnification of 96 k in counting mode and a pixel size of 0.840 Å at the specimen level with a dose rate of 0.92 e- per physical pixel per second, corresponding to 1.3 e- per Å² per second at the specimen level. The exposure time was 30.6 s, resulting in an accumulated dose of 40 e- per Å². Each movie includes 40 fractioned frames.

**Image processing.** All of the stacked frames were subjected to motion correction with MotionCor2[44]. Defocus was estimated using CTFFIND4[45]. A total of 153,004 particles were selected from 1701 micrographs using the EMAN2 suite (Supplementary Figs. 2 and 3a)[46]. The initial 3-D model was generated with 15,721 particles from 55 selected micrographs with underfocus values ranging between 2 and 3 μm using EMAN2. The particles were further analyzed with RELION3.0[47], and 123,728 particles were selected by 2-D classification and divided into four classes by 3-D classification resulting in only one good class containing 106,529 particles. The 3-D auto refinement without any imposed symmetry (C1) produced a map at 2.96 Å resolution after contrast transfer function refinement, Bayesian polishing, masking, and post-processing. Then, 1609 micrographs were further chosen based on the number of particles, the estimated image resolution, and the quality of Thon ring fitting. The selected 105,234 particle projections were subjected to subtraction of the detergent micelle density followed by 3-D auto refinement to yield the final map with a resolution of 2.82 Å according to the gold-standard Fourier shell correlation using a criterion of 0.143 (Supplementary Fig. 3)[48]. The local resolution maps were calculated on RESMAP[49].

**Model building and refinement of the LH1-RC complex.** The atomic model of the *Tch. tepidum* LH1-RC (PDB code 5Y5S) was fitted to the cryo-EM map obtained for the *Trv.* strain 970 LH1-RC using Chimera[50]. Amino acid substitutions and real space refinement for the peptides and cofactors were performed using COOT[51]. The C-terminal regions of the LH1 α-subunit were modeled ab-initio based on the density. The manually modified model was real space refined on PHENIX[52], and the COOT/PHENIX refinement was iterated until the refinements converged. The final model was further refined using REFMAC5 in the CCP-EM suite[53]. Finally, the statistics calculated using MolProbity[54] were checked. Figures were drawn with the Pymol Molecular Graphic System (Schrödinger)[55] and UCSF Chimera[50].

**Reporting summary.** Further information on research design is available in the Nature Research Reporting Summary linked to this article.

## Data availability

Map and model have been deposited in the EMDB and PDB with the accession codes: EMD-30314 and PDB-7C9R [https://doi.org/10.2210/pdb7C9R/pdb]. Other data are available from the corresponding authors upon reasonable request.

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

## Acknowledgements

We thank Anika Methner and Franziska Klann (Leibniz Institute DSMZ) for providing excellent technical assistance. This research was partially supported by Platform Project for Supporting Drug Discovery and Life Science Research (Basis for Supporting Innovative Drug Discovery and Life Science Research (BINDS)) from AMED under Grant Numbers JP20am0101118 (support number 1758) and JP20am0101116 (support number 1878), 17am0101116j0001, 18am0101116j0002, and 19am0101116j0003. R.K., M.H., and B.M.H. acknowledge the generous support of the Okinawa Institute of Science and Technology and the Japanese Cabinet Office. This work was supported in part by JSPS KAKENHI Grant Numbers JP16H04174, JP18H05153, JP20H05086, and JP20H02856, Takeda Science Foundation, and the Kurata Memorial Hitachi Science and Technology Foundation, Japan, and the National Key R&D Program of China (No. 2019YFA0904600).

## Author contributions

Z.-Y.W.-O. and K.T. designed the work, J.O. and M.T.M. provided materials and comments, K.T., R.K., Y.M., M.H., M.T., and M.I. performed the experiments, K.T., R.K., L.-J.Y., Y.K., A.M., B.M.H., and Z.-Y.W.-O. analyzed data, Z.-Y.W.-O. and K. T. wrote the manuscript.

## Competing interests

The authors declare no competing interests.
