## [Peer Review File · Nature Communications]

REVIEWER COMMENTS

Reviewer #1 (Remarks to the Author):

Review of Cryo-EM Structure of a Ca²⁺-Bound Photosynthetic LH1-RC Complex Containing Multiple $\alpha\beta$ -Polypeptides

By Kazutoshi Tani and coworkers

- What are the noteworthy results?

This an excellent paper that after some minor changes can be accepted by Nature Communications. The authors obtain and analyze Cryo-EM images plus modelling to solve the structure of the LH1-RC complex of the purple photosynthetic bacterium *Thiorhodovibrio* (Trv) strain 970. This complex is special because it displays the redmost absorption of a BChla containing purple bacterium, all the way up to 960 nm. This is due to the presence of 16 Ca²⁺ in the LH1 structure which tightly lock the $\alpha\beta$ -rings in a rigid configuration. The different α and β polypeptides occur in highly specified positions and with precise interactions with neighbouring polypeptides what suggests that the assembly of the complex is controlled by detailed and specific interactions between the rings and the RC in the center. The RC of Trv970 is at 920 nm, implying that trapping the energy from the antenna by the RC is an extremely uphill process.

- Will the work be of significance to the field and related fields? How does it compare to the established literature? If the work is not original, please provide relevant references.
The work is very original and strongly supports and further extends earlier work on bacterial reaction centers with BChl a as the major pigment. It further extends our ideas concerning the RC-LH1 complex of *Tch.tepidum* which has a similar ring of Ca²⁺ atoms linking the α - and β -rings. The organization of the different α - and β -polypeptides is a new new and surprising feature that hopefully will tell us in the future more of the assembly process of the RC-LH1 structure.

- Does the work support the conclusions and claims, or is additional evidence needed?

The work largely supports the conclusions and claims. I have a few minor comments that the authors could clarify

i) What precisely is the explanation for the LH1 redshift in *Tch.tepidum*? It would be good if this is explained in some detail because the authors us this as a model for the LH1 redshift in Trv. 970

ii) Has the uphill energy transfer from the LH1 antenna at 960 nm to P at 920 nm actually been observed. How efficient is it?

iii) On page 7 it is statedform an elliptical and partially overlapping ring..... What is that?

iv) ...plane of special pair...?

v) More extensive inhomogeneous narrowing of the BChla site energy. That should lead to a redshift plus a significant narrowing of the spectrum. Was that observed?

- Are there any flaws in the data analysis, interpretation and conclusions? - Do these prohibit publication or require revision?

No

- Is the methodology sound? Does the work meet the expected standards in your field?

Yes

- Is there enough detail provided in the methods for the work to be reproduced?

Yes

Rienk van Grondelle

Reviewer #2 (Remarks to the Author):

The article describes the cryo-EM Structure of a Ca²⁺-Bound Photosynthetic LH1-RC Complex Containing Multiple $\alpha\beta$ -Polypeptides.

The researchers of the reviewed article have extensional expertise in the described subject (Imanishi et al., 2019).

Previously the dual Role for Ca²⁺ impact on the Spectral Diversity and Stability of Light-Harvesting 1 Reaction Center Photocomplexes of Purple Phototrophic Bacteria was assumed (O. Rucker et al., Archives of Microbiology, 2012). By the same study was proposed that the close proximity localization of side chain of the alfa His to the BChla exerts a modulating effect in the spectral properties of the highly unusual LHC1 complex of strain 970 (O. Rucker et al., Archives of Microbiology, 2012).

The authors of the reviewed article have successfully revealed and studied the structure of the unique purple phototrophic bacterium *Thiorhodovibrio* (Trv.) strain 970 using the cryo-EM technique at 2.82 Å resolution. The authors locate and describe the arrangement of the Reaction Center subunits, the LH1 antenna complex, with 16 repeating single transmembrane α and β helix pairs, the correspondent cofactors, and the location of Ca²⁺ ions. They also compared the obtained structure with another thermophilic purple bacterium *Thermochromatium tepidum* resolved by x-ray to 1.9 Å.

It was shown that despite non-thermophilic nature, Trv. strain 970 possesses the largest redshift absorbance among LH1 complexes reported. This redshift allows the studied purple bacterium to be beneficial in growth under different environments conditions.

Overall, the performed study is of high importance due to the quality and the variety of techniques used together with the obtained reliable results.

I believe the reviewed article is of interest as for the photosynthetic /biophysical community as for the applied scientific studies.

I am assured that the article can be published after several corrections applied.

My main concerns are related to some inaccuracies found in the text and the and some inconsistency in the illustrations for the main and supplementary figures.

Text

Abstract

1. Page 5 -- As a consequence, the LH1 inner and outer rings form a tight network connected by Ca²⁺.

That is the first time that the arrangement of LH1 is mentioned. It is not explained that LH1 consists of α (inner) and β (outer) helix subunits.

2. No theory is given for the arrangement of the reaction center of the *Thiorhodovibrio* (Trv.) strain 970.

3. Page 9 -- One α 2- and one α 4-polypeptide were identified in the cryo-EM structure (Figure 3a and 3b).

In Figure 3a and 3b we see no cryo-EM structural maps. Instead, we see the cartoon structural modes of RC-LH1 (3a) and LH1 (3b) both in top view, probably based on the fitted model into the cryo-EM map.

Figures

Figure 1. Overall structure and cofactor arrangement of the Trv. strain 970 LH1-RC complex – where the correspondent model is illustrated.

- The color scheme listed at the end of the description for Figure 1 represents the colors of the subunits as LH1 α , green; LH1 β , slate blue; L-subunit, magenta; M-subunit, blue; Ca²⁺, orange ball; BChl a, red sticks; 3,4,3',4'-tetrahydrospirilloxanthin, yellow sticks; DDM, black sticks. Phospholipids are omitted for clarity. Cofactors in L- and M-subunits of RC are coloured in magenta and blue, respectively.

At the same time, the color description of the RC subunits: C-subunit (cyan) and the cytoplasmic H-subunit (orange) are given separately from the color scheme in Figure 1a.

That can be not very clear for the reader.

- The color for the H-subunit (orange) is the same as for the Ca²⁺ ions – it is not critical but can be misleading.

Figure 3. Arrangement of the LH1 multiple polypeptides.

- In Figure 3a the reaction center is shown in the form of the gray sticks without specification of each of the subunits. I would propose to color the reaction center subunits in the same color as in Figure 1a; 1b and keep them transparent as in Figure 3c.

- In Figure 3c the reaction center subunits are colored as L- subunit, orange; M-subunit, blue; C-subunit – magenta. This color scheme differs from the one used in Figure 1a; 2b.

I find it confusing for the reader and propose to keep the same coloring of the main structures of the LH1-RC complex within the article.

Supplementary Figure 2d.

- Please add the Corrected/unmasked/phase randomized FSC plots for your final cryo-EM map.

Recommendations:

1. Make a figure of cryo-EM density map fitted in the model.
2. To evaluate the angular coverage of the projection sphere, please add to Supplementary Figure 2 the picture with an angular distribution plot for the final reconstruction.
3. To evaluate the local resolution, please add the picture to the Supplementary Figure 2 of the central slice projection showing the local resolution distribution within the cryo-EM map.

Sincerely,

Dr. Dmitry A. Semchonok, Ph.D.

e-mail: dmitry.semchonok@bct.uni-halle.de

Response to reviewers:

Reviewer #1

Reviewer #1's comments: Point 1

This an excellent paper that after some minor changes can be accepted by Nature Communications. The authors obtain and analyze Cryo-EM images plus modelling to solve the structure of the LH1-RC complex of the purple photosynthetic bacterium *Thiorhodovibrio (Trv)* strain 970. This complex is special because it displays the redmost absorption of a BChl_a containing purple bacterium, all the way up to 960 nm. This is due to the presence of 16 Ca²⁺ in the LH1 structure which tightly lock the ab-rings in a rigid configuration. The different a and b polypeptides occur in highly specified positions and with precise interactions with neighbouring polypeptides what suggests that the assembly of the complex is controlled by detailed and specific interactions between the rings and the RC in the center. The RC of Trv970 is at 920 nm, implying that trapping the energy from the antenna by the RC is an extremely uphill process.

- Will the work be of significance to the field and related fields? How does it compare to the established literature? If the work is not original, please provide relevant references.

The work is very original and strongly supports and further extends earlier work on bacterial reaction centers with BChl a as the major pigment. It further extends our ideas concerning the RC-LH1 complex of *Tch. tepidum* which has a similar ring of Ca²⁺ atoms linking the a- and b-rings. The organization of the different a- and b-polypeptides is a new and surprising feature that hopefully will tell us in the future more of the assembly process of the RC-LH1 structure.

- Does the work support the conclusions and claims, or is additional evidence needed?

The work largely supports the conclusions and claims. I have a few minor comments that the authors could clarify

i) What precisely is the explanation for the LH1 redshift in *Tch. tepidum*? It would be good if this is explained in some detail because the authors use this as a model for the LH1 redshift in *Trv. 970*

Our response:

We appreciate the reviewer's positive assessment of our work. The origin of the *Tch. tepidum* LH1 redshift has been extensively investigated by Ma and co-workers (Ref. 14, 15, 29, 30) and our group (Ref. 7–10, 18–20) using both spectroscopic and biochemical analyses as described in the text and cited in the references. For an easy browse, we have reorganized these references and put them together in the sentence that mentioned on this point (first paragraph in the Discussion section).

Reviewer #1's comments: Point 2

ii) Has the uphill energy transfer from the LH1 antenna at 960 nm to P at 920 nm actually been observed. How efficient is it?

Our response:

The energy transfer from the *Trv. 970* LH1 at 960 nm to P in RC at 920 nm has been measured by Permentier, et al. (Ref. 6), yielding a time constant of 65 ps. We have added a sentence to mention this in the Introduction section. Although no efficiency has been reported,

a preliminary result from our own group (unpublished) showed extremely low fluorescence in this process, indicating a highly efficient energy transfer despite its uphill nature.

Reviewer #1's comments: Point 3

iii) On page 7 it is stated ...form an elliptical and partially overlapping ring..... What is that?

Our response:

We have modified this sentence to clarify the BChl *a* organization in the LH1 ring.

Reviewer #1's comments: Point 4

iv)...plane of special pair...?

Our response:

We have modified this sentence to clarify the orientation of the special pair BChls *a*.

Reviewer #1's comments: Point 5

v) More extensive inhomogeneous narrowing of the BChl *a* site energy. That should lead to a redshift plus a significant narrowing of the spectrum. Was that observed?

Our response:

The width of the *Trv.* 970 LH1-Qy band is slightly larger than that of *Tch. tepidum* at room temperature. Therefore, the description in the old version was inaccurate, and we have removed the corresponding sentence in the revised manuscript (second paragraph in Discussion section). We thank the reviewer for pointing out this issue.

Reviewer #2

Reviewer #2's comments: Point 1

The researchers of the reviewed article have extensional expertise in the described subject (Imanishi et al., 2019).

Previously the dual Role for Ca²⁺ impact on the Spectral Diversity and Stability of Light-Harvesting 1 Reaction Center Photocomplexes of Purple Phototrophic Bacteria was assumed (O. Rucker et al., Archives of Microbiology, 2012). By the same study was proposed that the close proximity localization of side chain of the alfa-His to the BChl *a* exerts a modulating effect in the spectral properties of the highly unusual LHC1 complex of strain 970 (O. Rucker et al., Archives of Microbiology, 2012).

The authors of the reviewed article have successfully revealed and studied the structure of the unique purple phototrophic bacterium Thiorhodovibrio (*Trv.*) strain 970 using the cryo-EM technique at 2.82 Å resolution. The authors locate and describe the arrangement of the Reaction Center subunits, the LH1 antenna complex, with 16 repeating single transmembrane α and β helix pairs, the correspondent

cofactors, and the location of Ca^{2+} ions. They also compared the obtained structure with another thermophilic purple bacterium *Thermochromatium tepidum* resolved by x-ray to 1.9 Å. It was shown that despite non-thermophilic nature, *Trv.* strain 970 possesses the largest redshift absorbance among LH1 complexes reported. This redshift allows the studied purple bacterium to be beneficial in growth under different environments conditions.

Overall, the performed study is of high importance due to the quality and the variety of techniques used together with the obtained reliable results.

I believe the reviewed article is of interest as for the photosynthetic /biophysical community as for the applied scientific studies.

I am assured that the article can be published after several corrections applied.

My main concerns are related to some inaccuracies found in the text and the and some inconsistency in the illustrations for the main and supplementary figures.

Text

Abstract

1. Page 5 -- As a consequence, the LH1 inner and outer rings form a tight network connected by Ca^{2+} . That is the first time that the arrangement of LH1 is mentioned. It is not explained that LH1 consists of α (inner) and β (outer) helix subunits.

Our response:

We thank the reviewer for the positive assessment of our work. We have added an explanation on the arrangement of α (inner) and β (outer) polypeptides as their first appearance (first paragraph on Page 5).

Reviewer #2's comments: Point 2

2. No theory is given for the arrangement of the reaction center of the *Thiorhodovibrio (Trv.)* strain 970.

Our response:

We have added a description on the structure of the *Trv.* strain 970 RC (first paragraph in the Result section) and Supplementary Figure 7 for detailed comparisons on each subunit of the RCs between *Trv.* strain 970 and *Tch. tepidum*.

Reviewer #2's comments: Point 3

3. Page 9 -- One α_2 - and one α_4 -polypeptide were identified in the cryo-EM structure (Figure 3a and 3b).

In Figure 3a and 3b we see no cryo-EM structural maps. Instead, we see the cartoon structural modes of RC-LH1 (3a) and LH1 (3b) both in top view, probably based on the fitted model into the cryo-EM map.

Our response:

We have added cryo-EM structural maps and the fitted models (Supplementary Figure 5) to show that the structures of α_2 - and α_4 -polypeptides were derived from the density maps.

Reviewer #2's comments: Point 4

Figure 1. Overall structure and cofactor arrangement of the *Trv.* strain 970 LH1-RC complex – where the correspondent model is illustrated.

- The color scheme listed at the end of the description for Figure 1 represents the colors of the subunits as LH1 α , green; LH1 β , slate blue; L-subunit, magenta; M-subunit, blue; Ca^{2+} , orange ball; BChl *a*, red sticks; 3,4,3',4'-tetrahydrospirilloxanthin, yellow sticks; DDM, black sticks. Phospholipids are omitted for clarity. Cofactors in L- and M-subunits of RC are coloured in magenta and blue, respectively.

At the same time, the color description of the RC subunits: C-subunit (cyan) and the cytoplasmic H-subunit (orange) are given separately from the color scheme in Figure 1a.

That can be not very clear for the reader.

- The color for the H-subunit (orange) is the same as for the Ca^{2+} ions – it is not critical but can be misleading.

Our response:

- We have modified the color description on the RC subunits in Figure 1 caption according to the reviewer's suggestion.
- The color for the RC H-subunit is orange, but the color for Ca^{2+} was actually gold. We have changed the description for the Ca^{2+} in Figure 1 caption, and thank the reviewer for pointing out this issue.

Reviewer #2's comments: Point 5

Figure 3. Arrangement of the LH1 multiple polypeptides.

- In Figure 3a the reaction center is shown in the form of the gray sticks without specification of each of the subunits. I would propose to color the reaction center subunits in the same color as in Figure 1a; 1b and keep them transparent as in Figure 3c.

- In Figure 3c the reaction center subunits are colored as L- subunit, orange; M-subunit, blue; C-subunit – magenta. This color scheme differs from the one used in Figure 1a; 2b.

I find it confusing for the reader and propose to keep the same coloring of the main structures of the LH1-RC complex within the article.

Our response:

- While Figure 1 shows overall structure of the *Trv.* 970 LH1-RC including all proteins and cofactors, Figure 3a was intended to show up the organization of the LH1-only polypeptides to emphasize the unique arrangement of the multiple $\alpha\beta$ polypeptides. In order to get maximum contrast, we prefer all of the RC subunits to be shown in gray cartoons after discussion with our co-authors. According to the reviewer's suggestion, we have added Supplementary Figure 5 to show the individual RC subunits in different colors inside the LH1 ring.
- Due to the limited number of simple colors available and in order to get maximum contrast for emphasizing the interactions of $\alpha 2$ -L(RC) and $\alpha 4$ -M(RC), the colors used for RC L- and M-subunits in Figure 3c are different from those in Figure 1. To make this clear for avoiding confusion, we have added a note in the figure legend of Figure 3c.

Reviewer #2's comments: Point 6

Supplementary Figure 2d.

- Please add the Corrected/unmasked/phase randomized FSC plots for your final cryo-EM map.

Our response:

We have added these plots in Supplementary Figure 3b.

Reviewer #2's comments: Point 7

Recommendations:

1. Make a figure of cryo-EM density map fitted in the model.
2. To evaluate the angular coverage of the projection sphere, please add to Supplementary Figure 2 the picture with an angular distribution plot for the final reconstruction.
3. To evaluate the local resolution, please add the picture to the Supplementary Figure 2 of the central slice projection showing the local resolution distribution within the cryo-EM map.

Our response:

- We have added Supplementary Figure 5 to show the cryo-EM density map with fitted model.
- We have added Supplementary Figure 3c to show the angular distribution for the final reconstruction.
- We have added Supplementary Figure 4 to show the central slice projection for the local resolution distribution within the cryo-EM map.